# Demographic and COVID Experience Predictors of COVID-19 Risk Perception among Chinese Residents in Canada

**DOI:** 10.3390/ijerph192114448

**Published:** 2022-11-04

**Authors:** Mariah Lecompte, Alyssa Counsell, Lixia Yang

**Affiliations:** Department of Psychology, Toronto Metropolitan University, Toronto, ON M5B 2K3, Canada

**Keywords:** COVID-19, demographic predictors, risk perception, Chinese residents in Canada

## Abstract

The current study aimed to identify demographic and COVID experience predictors for COVID-19 risk perception among Chinese residents in Canada. A final sample of 653 participants aged 18 and up completed an online survey in simplified Chinese during the peak of the first wave of COVID-19 (25 April–10 June 2020). After removing those with missing data on demographic covariates, as missing data cannot be imputed, 444 were included in the structural equation model, and COVID-19 risk perception was indexed by three outcome variables: self-infection risk perception (i.e., likelihood of personal infection of COVID-19); threat perception (i.e., whether the pandemic is a real threat); and future infection rate prediction (i.e., a latent variable for community, Ontario, Canada, and World infection rate predictions). Predictors included demographic (i.e., income, gender, education, age, household size, employment status, and life satisfaction) and COVID experience variables (i.e., personal connection with confirmed or suspected COVID-19 cases, self-isolation experience, perceived anti-Chinese discrimination, and confusion over COVID-19 information). In the structural equation model, we found increased risk perception for the following demographic and COVID experience predictors; women, relatively higher education, living alone, working in a medical field, lower in life satisfaction, having personal connection with confirmed or suspected COVID-19 cases, with perceived anti-Chinese discrimination, or showing high confusion over COVID-19 information.

## 1. Introduction

On 11 March 2020, the World Health Organization declared COVID-19 a global pandemic [1]. During the first 6 weeks of quarantine, a higher perceived COVID-19 risk was associated with greater depressive symptoms, suggesting that those who viewed COVID-19 as a more tangible concern were also experiencing more emotional distress [2]. Specifically, more anxiety and depression symptoms were associated with those experiencing a high COVID-19 risk perception [3]. Furthermore, attending to COVID-19-related information and perceived severity (i.e., threat perception) of the pandemic resulted in greater preventative behavior endorsement and risk awareness [4,5]. Additionally, those with a personal experience with COVID-19 tend to perceive COVID-19 as a greater risk than those without direct experience [6]. Thus, overall, attending to COVID-19-related news and having a personal experience with COVID-19 increases perceived COVID-19 risk, which is in turn associated with greater distress.

In addition to the relationship between COVID-19 risk perception and psychological distress [2,3,6,7], previous work has also identified demographic predictors (e.g., being a woman, in young age, having lower income, or fewer years of education) for psychological distress during the COVID-19 pandemic [3,8,9,10,11,12]. However, the direct prediction of demographic and COVID experience variables for COVID-19 risk perception remains unclear. This question, however, is important for developing pandemic-coping measures or programs appropriate for the target populations with high-risk perception as stratified by a specific demographic or pandemic experience profiles. 

It should be noted that although COVID-19 risk perception has been measured in previous studies [2,3,6,7], the results are not directly comparable considering the measurements are different across studies. For instance, Kim (2020) had participants report if they were at a lower, equal, or greater risk of becoming infected than others, while Hyland et al. (2020) used a 10-point Likert scale for perceived self-infection risk. Xin et al. (2020) used temporally based questions, asking perception of infection risk during the next year, with a 5-point Likert scale. Dryhurst et al. (2020) assessed risk perception with a more holistic approach, creating an index that covered likelihood, worry, and temporal-spatial risk perception, based on 7-point Likert scales. In light of these discrepancies in measurement, the current study adopted a more comprehensive approach to measure risk perception using three types of risk-perception questions: self-infection risk, threat perception, and future infection rate prediction. This approach allowed for different types of risk-perception to be analyzed separately in the same model, as they may have different relationships with the predictors. Additionally, the utilization of a percentage prediction open response format for the future risk prediction provided participants an opportunity to openly estimate and report their perception for future infection rate at different scales (i.e., local community, Ontario, Canada, and World). Finally, it allowed us to address not only self-infection but also broader community or global infection rate prediction. 

The current study focused on Chinese residents in Canada. Compared to other Canadians, it has been reported that Canadian Chinese communities took timely implementations of self-protection measures to combat COVID-19 at the very beginning of the pandemic [13]. In fact, reports suggest that Chinese communities adopted self-isolation and mask-wearing measures even before these behavior protocols became mandatory [13]. This early adoption of protective methods suggests that Chinese communities may have a higher sensitivity to COVID-19 risk perception.

In light of these previous findings, this study aimed to identify critical demographic and COVID experience predictors for COVID-19 risk prediction. Specifically, we sought to determine the demographic profiles and COVID experiences that would make individuals more vulnerable to a pessimistic risk prediction of the pandemic among Chinese residents in Canada.

## 2. Materials and Methods

### 2.1. Sample 

Using a snowball recruitment approach, we recruited a sample of 656 Chinese residents in Canada aged 18 and over through WeChat and the Internet to participate in this study. The final sample included 653 participants, excluding three cases who were previously diagnosed with COVID-19 (considering the current analysis primarily focused on risk perception, including risk of self-infection). For the structural equation model, 444 participants were included after removing all cases with missing values on the demographic covariates, as missing data cannot be imputed for demographic information (*n* = 209). As displayed in Table 1, the sample included 31% with an average income, 51% women, 39% with university or college education, 56% aged between 35 and 64, 38% living with three to four people, 43% working in a non-medical field, 71% without loved ones contracting COVID-19, 50% having self-isolation experience, 30% with perceived anti-Chinese discrimination, and 39% feeling confused about COVID-19 information.

### 2.2. Survey

The online survey was built in QualtricsTM and delivered in simplified Chinese. The data were collected at the peak of the first wave of the COVID-19 pandemic in Canada (i.e., 25 April to 10 June 2020). The survey included questions on demographic predictors, COVID experience predictors, and COVID-19 risk perception. The current manuscript specifically examines the predictions of demographic and COVID experience variables for COVID-19 risk perception, as indexed by self-infection risk, threat perception, and future infection rate prediction. The novelty and unique value of this study stems from the focus on the currently under-researched Chinese Canadians, refined evaluation of different types of risk prediction (e.g., percentile infection rate prediction), and the consideration of comprehensive potential demographic and COVID experience predictors. 

### 2.3. The Structural Equation Model (SEM)

We constructed a SEM (Figure 1 in Section 3) that included critical demographic (i.e., income, gender, education, age, household size, employment status, and life satisfaction) and COVID experience variables (i.e., personal connection with confirmed or suspected COVID-19 cases, with self-isolation experience, perceived anti-Chinese discrimination, and confusion over COVID-19 information) as predictors for risk perception, operationalized as the prediction of self-infection of the virus (i.e., self-infection risk), pandemic as a threat (i.e., threat perception), and the population future infection rate (i.e., future infection rate prediction). We hypothesized that lower income, being a woman, lower education, younger adults, a smaller household size, and lower life-satisfaction would be differentially related to greater risk prediction. We also predicted that COVID experience (i.e., personal connection with confirmed or suspected cases, self-isolation experience, perceived anti-Chinese discrimination, and confusion over COVID-19 information) will also be related to higher risk predictions. 

#### 2.3.1. Demographic Predictors

The main demographic predictors included in the model are outlined in Table 1. Education was categorically recoded into three groups: “0” for those with high school, technical secondary school, or less, which was used as the reference group, “1” for those with university or college education, and “2” for those with a Master’s degree or higher. Age was also recategorized in order to collapse the multiple levels into three meaningful age groups: “0” for young adults aged 18–34, which was used as the reference group, “1” for middle adulthood aged 35–64, and “2” for older adults aged 65 and over. Household size was recategorized: “0” for those who live alone, “1” for those households of two people, “2” for those households of three to four people, and “3” for households of five or more people. Finally, the employment type was categorically recoded into three groups deemed most interesting for this analysis: “0” for those working in a medical field (i.e., doctor, nurse), “1” for those working in a non-medical field (i.e., employed other careers, contract/part-time, self-employed), and “2” for those not currently employed (i.e., full-time student, retired, laid off, stay at home caregiver). Life satisfaction was calculated with the sum score on The Satisfaction with Life Scale (SWLS), a 5-item questionnaire with a 7-point Likert scale (*M* = 24.78, *SD* = 5.819 in our sample) [14]. The “code used in analysis” column shows how each of the variables were coded in the model. 

#### 2.3.2. COVID Experience Predictors

Personal connection with confirmed or suspected COVID-19 cases was assessed with the question “Are there any confirmed or suspected COVID-19 cases in your family/friends/colleagues?”, to which the responses showed a diverse distribution and were thus categorically recoded as “1” (i.e., “NO”) or “2” (i.e., “CONFIRMED”, “SUSPECTED” or “NOT SURE”). Self-isolation experience was assessed as one of the behaviours possibly experienced during the pandemic, coded as “YES” if it was selected, and “No” if not selected. Perceived anti-Chinese discrimination (i.e., “Do you think there is a rising discrimination against Chinese during the pandemic?”) and confusion over COVID-19 information (i.e., “to which degree you feel confused over COVID-19 information”) were rated on a 5-point Likert-scale (i.e., from “completely disagree” to “completely agree” for the discrimination item; from “never” to “always” for the information confusion item). 

#### 2.3.3. Outcome Variables 

Table 2 displays the outcome variables and their associated questions and coding. Self-infection and threat perception were each assessed with one question based on a 5-point Likert scale. A latent variable was created for the future infection rate prediction, integrating four questions requesting an estimate of the percentage of the population that will be infected in the future in local community, Ontario, Canada, and World. This combination was a valid approach considering two things: (1) the lack of a positivity bias for personal over collective future prediction in this sample, similar to previous findings with Chinese samples [15,16]; (2) all four items loaded (standardized values ranged from 0.801 to 0.965) well above the recommended value of 0.40 [17], supporting the appropriateness of a single latent variable for future infection prediction [18,19]. Additionally, all four variables showed consistent positive correlations (*rs* = 0.636–0.888, *ps* < 0.001). Considering the large scale of this latent variable (i.e., 1–100) relative to the other predictors in the model, we divided the values by 10 to minimize possible model nonconvergence due to variance discrepancies. Any ratings larger than 100% were removed from the final analysis.

## 3. Results

### 3.1. Model Fit 

We used three indices to measure the model’s goodness of fit: Standardized Root Mean Square Residual (SRMR) with a score of 0.08 or less, Root Mean Square Error of Approximation (RMSEA) with a score of 0.06 or less, and Comparative Fit Index (CFI) with a score of over 0.95 [20,21]. Model fit statistics were as follows, χ² (56) = 95.82, *p* ≤ 0.001, SRMR = 0.015, RMSEA = 0.040, CFI = 0.973, suggesting an acceptable model fit. 

### 3.2. Model Assumptions

The multicollinearity assumption was not violated for the model, as none of the variables in our sample had variance inflation of over 2. Additionally, the assumption of linearity was not violated, as determined through a scatterplot matrix. The full information maximum likelihood (FIML) method was used to preserve data where possible, but due to missingness on demographic covariates, 209 participants were removed from the final analysis [20,22]. An investigation of an individual histogram for each variable, including outcome and predictor variables, showed that the assumption of normality was slightly violated. Further investigation of the multivariate normality using the mvn function in the MVN R package [23] showed a kurtosis score of 11.78, suggesting a violation (kurtosis score > 10). In order to adjust for this violation of the normality assumption, we used the robust MLR method for model estimation [24]. All reported statistics from the structural equation model, including the model fit statistics above, are from the MLR estimated model. Table 3 displays the correlations between all the variables included in the model. 

### 3.3. Structural Equation Model (SEM) for COVID-19 Risk Perception 

Figure 1 includes a path diagram of the relationships estimated with the standardized coefficients. Only significant estimates are reported numerically (represented by the solid black lines). We display standardized regression effect size estimates to compare the relative magnitude of effects [25]. The model identified the following significant predictors for each outcome variable: (1) working in a medical field, life satisfaction, personal connection with COVID-19, and perceived discrimination, for self-infection prediction; (2) household size, personal connection with COVID-19, and perceived discrimination, for threat perception; (3) being a woman, higher education, personal connection with COVID-19, and perceived anti-Chinese discrimination, and COVID-19 information confusion for future infection rate prediction. As gender, education, household size, employment, and a personal connection with COVID-19 were all binary predictors, and thus not interpretable in regards to a 1 SD unit change, these results will be discussed using the partially standardized coefficients [26]. All remaining results will be discussed using standardized coefficients.

**Figure 1 ijerph-19-14448-f001:**
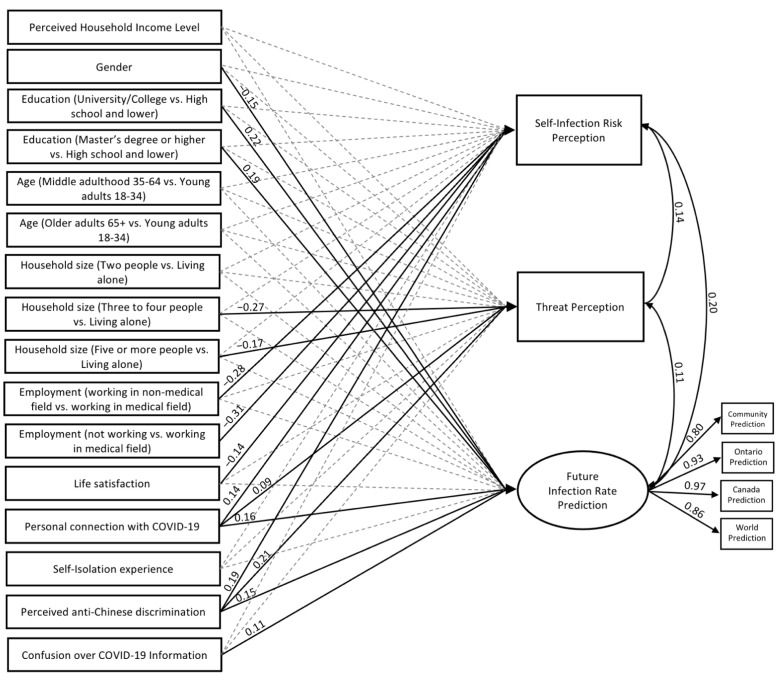
The SEM path diagram. Note: All presented coefficients are standardized. Basic model fit: χ² (56) = 95.82, *p* ≤ 0.001, SRMR = 0.015, RMSEA = 0.040, CFI = 0.973.

#### 3.3.1. Self-infection Risk Prediction

After controlling for other predictors, the results identified the following demographic and COVID experience predictors for self-infection risk prediction (R^2^ = 0.161): compared to working in a medical field, those unemployed scored lower by 0.659 SD units (*p* = 0.025), and those working in a non-medical field scored lower by 0.571 SD units (*p* = 0.043), on self-infection risk prediction. For life-satisfaction, a 1 SD unit change in life satisfaction predicted a −0.144 SD unit change (*p* = 0.004) in self-infection risk prediction. Moreover, those with personal connection with confirmed or suspected COVID-19 cases scored higher by 0.324 SD units (*p* = 0.002); and a 1 SD unit change in perceived discrimination predicted a 0.194 SD unit change (*p* < 0.001) in self-infection risk prediction. 

#### 3.3.2. Threat Prediction

After controlling for other predictors, the results identified the following demographic and COVID experience predictors for threat prediction (R^2^ = 0.140): compared to those living alone, households of 3 to 4 people scored lower by 0.529 SD units (*p* = 0.002), and households of 5 or more people scored lower by 0.423 SD units (*p* = 0.030), on threat perception. Moreover, those with a personal connection with confirmed or suspected COVID-19 cases scored higher by 0.207 SD units (*p* = 0.029), and a 1 SD unit change in perceived discrimination predicted a 0.212 SD unit change (*p* < 0.001), in threat perception. 

#### 3.3.3. Future Infection Rate Prediction

After controlling for other predictors, the results identified the following demographic and COVID experience predictors for future infection rate prediction (R^2^ = 0.155): women scored 4.31% higher on future infection rate prediction than men (*p* = 0.005). Compared to those with high school education or less, those with university or college level education scored 5.77% higher (*p* = 0.007), and those with a master’s degree or higher education scored 5.20% higher (*p* = 0.035). Moreover, those with personal connection with confirmed or suspected COVID-19 cases scored higher by 4.89% (*p* = 0.005). A 1 SD unit change in perceived discrimination predicted a 0.149 SD unit change (*p* = 0.006), and a 1 SD unit change in confusion in COVID-19 information predicted a 0.113 SD unit change (*p* = 0.031), in future infection rate predictions.

## 4. Discussion

The results of this study suggest that during the peak of the first wave of the pandemic, the pandemic risk perception among Chinese residents in Canada was related to gender, education, household size, employment in a medical field, life satisfaction, a personal connection with confirmed or suspected COVID-19 cases, perceived discrimination, and COVID-19 information confusion. It should be noted that all the standardized effects are in the smaller range [25]. Additionally, the R-squared values were considered weak for self-infection risk (i.e., *R^2^* = 0.161), threat perception (i.e., *R^2^* = 0.140), and future infection rate predictions (i.e., *R^2^* = 0.155) [27]. These small effects may be because the perception of COVID-19 risk was not fully captured by the predictors in this model. Other factors that might predict risk perception (e.g., the type and method that health messages were consumed [28]) were unfortunately not captured in the current study. 

### 4.1. Demographic Predictors

The results identified women as a significant predictor for higher future community infection rate of COVID-19, although no gender differences were identified for self-infection or threat perception. Women’s pessimistic outlook on the future infection rate might be related to their differentially higher level of emotional distress during the pandemic compared to men. For example, it has been found that during the pandemic, women experienced a higher level of stress, anxiety, and depression compared to men [9,10] and those with a higher perceived COVID-19 risk are associated with more depressive symptoms [2]. This finding might also be due to the traditional family and gender expectations that are still widespread in Chinese communities, where the caregiver role is typically attributed to women [29,30]. As such, COVID-19 may place additional expectations on women in caregiver roles, as they now must balance the new emotional and social needs of their family, as well as their own. These additional expectations placed on women for the care of the families could conceivably lead to higher levels of stress and emotional distress [9,10]. Thus, the emotional distress and the society-oriented care-giving role might make women to hold a more pessimistic future outlook for the infection rate. 

Additionally, when compared to having a high school or lower education, those with university, college, master’s degree or higher predicted higher future infection rates, potentially because those with higher education might pay attention to information from more authorized channels and thus might be more likely to have some realistic and scientific understanding of infection severity and long-term impacts of COVID-19. 

In regards to household size, the results showed that in comparison to those living alone, households containing three or more people showed a lower threat perception, potentially because the social support or connection in larger households may minimize their threat perception. The results also revealed that working in the medical field predicted an increase in self-infection perception, probably due to their increased exposure risk as a result of their employment. Finally, higher life satisfaction predicted lower self-infection perception, suggesting that higher life satisfaction (which might be related to higher social economic status and better psychological wellbeing) serves as a buffer against self-infection anxiety, because they have sufficient resources to deal with it if they do get infected. This finding supports previous research of the inverse relationship between life satisfaction and psychological distress [31,32].

### 4.2. COVID Experience Predictors

Across the board, personal connection with confirmed or suspected COVID-19 cases, and perceived discrimination, consistently predicted higher risk prediction across all three levels. This finding suggests that these predictors are robust and sensitive risk factors for COVID-19 risk perception, especially among Chinese living in Canada. There is also a possibility that individuals who are perceiving discrimination, or who have a personal connection to COVID-related cases, are experiencing higher levels of distress. In particular, worry about personally contracting COVID-19 has been significantly associated with COVID-19 peritraumatic distress [33]. Similarly, previous research showed that perceived and experienced racial discrimination serve as robust predictors for mental health condition among Chinese living in Canada [34,35]. 

With regards to COVID-19 information confusion, higher confusion predicted higher future infection rate prediction. This finding suggests that those who felt that they understood the information they received about the pandemic viewed the pandemic as less of a threat than those who did not understand. What is unclear however, is what was causing an increase in confusion. Potentially those who felt confused might have attended to multiple (reliable or unreliable) sources of COVID-19 information, and may have been receiving conflicting information. This may increase their distress and uncertainty, and thus make them likely to perceive the pandemic as more threating [33].

### 4.3. Limitations

A major limitation of this study is the lack of diversity in the sample. The majority of our respondents had an average income, were women, were university or college educated, were 35–64 years old, lived with three to four people, and worked in a non-medical field. The method of recruitment, the snowball sampling approach, may partly contribute to the lack of diversity in the sample. Follow-up research would be needed to diversify the sample and enhance the generalizability of the results. Additionally, it should be noted that there was a lot of missing demographic information, particularly with undisclosed gender. The survey itself is limited and does not capture other potentially important predictive variables for COVID-19 risk perception.

## 5. Conclusions

In conclusion, the results identified several significant predictors of COVID risk perception among Chinese residents in Canada. Consistent with previous findings, we found that being a woman predicted a higher future infection rate, probably due to their higher psychological distress and heavier caregiver social roles [9,10,11,12]. A larger household size was generally related to a lower threat perception relative to living alone, while a higher education was related to a higher future infection rate predication. Additionally, working in a medical field, and decreased life satisfaction appears to predict higher self-infection perception. Most importantly, having personal connection with COVID-19 and perceived anti-Chinese discrimination was associated with higher COVID-19 risk perception across all three levels [6,34]. Finally, those with greater confusion over COVID-19 information predicted a higher future infection rate. The results remain after controlling for all other covariates. Future research may examine how these predictors are related to long-term preventative behavior measure adherence and vaccine turnout rates among Chinese residents in Canada in comparison to other population groups. 

## Figures and Tables

**Table 1 ijerph-19-14448-t001:** Predictors included in the model (*n* = 653).

Predictor Category	Predictors		Response Rate: *n (%)*	Code
Demographic	Perceived Household Income Level	Low	55 (8.42%)	1
		Lower than average	67 (10.26%)	2
		Average	202 (30.93%)	3
		Higher than average	118 (18.07%)	4
		High	17 (2.60%)	5
	Gender	Women	334 (51.15%)	1
		Men	124 (18.99%)	2
	Education	High school/secondary school or less	38 (5.82%)	0
		University/College	256 (39.20%)	1
		Master’s degree or higher	167 (25.57%)	2
	Age	Young adults (18–34)	44 (6.74%)	0
		Middle adulthood (35–64)	364 (55.74%)	1
		Older adults (65+)	54 (8.27%)	2
	Household Size	Alone (1 person)	22 (3.37%)	0
		2 people	98 (15.01%)	1
		3–4 people	249 (38.13%)	2
		5 or more	85 (13.02%)	3
	Employment Status	Working in a medical field	18 (2.76%)	0
		Working (non-medical field)	284 (43.49%)	1
		Not currently employed	158 (24.20%)	2
COVID Experience	Personal Connection with COVID-19	No	466 (71.36%)	1
		Yes (Confirmed/Suspected/Unsure)	147 (22.51%)	2
	Self-isolation experience	No/Not selected Yes	285 (43.64%)	1
			324 (49.62%)	2
	Perceived Discrimination	Completely disagree	10 (1.53%)	1
		Somewhat disagree	62 (9.49%)	2
		Neutral	172 (26.34%)	3
		Somewhat agree	198 (30.32%)	4
		Completely agree	85 (13.02%)	5
	Confusion over COVID-19 Information	Never	43 (6.58%)	1
		Seldom	98 (15.01%)	2
		Sometimes	255 (39.05%)	3
		Often	114 (17.46%)	4
		Always	21 (3.22%)	5

**Table 2 ijerph-19-14448-t002:** Outcome variables’ breakdown in the model (*n* = 653).

Outcome Variables	Survey Question	Likert Scale	Response Rate *n (%)*
Self-infection Risk Perception	How likely do you think it is that you will be infected with the COVID-19?	1 = Very unlikely	28 (4.28%)
		2 = Unlikely	124 (18.99%)
		3 = Neutral	290 (44.41%)
		4 = Likely	132 (20.21%)
		5 = Very likely	30 (4.59%)
Threat Perception	Do you believe that COVID-19 pandemic is a real threat?	1 = Completely disagree	3 (0.46%)
		2 = Somewhat disagree	17 (2.60%)
		3 = Neutral	69 (10.57%)
		4 = Somewhat agree	229 (35.07%)
		5 = Completely agree	212 (32.47%)
Future Infection Rate Prediction	Please estimate the percentage of your community population will be infected?	Percentage (%) input	419 (64.17%)
	Please estimate the percentage of Ontario population will be infected?	Percentage (%) input	412 (63.09%)
	Please estimate the percentage of Canadian population will be infected?	Percentage (%) input	398 (60.95%)
	Please estimate the percentage of World population will be infected?	Percentage (%) input	401 (61.41%)

**Table 3 ijerph-19-14448-t003:** Correlations between predictor and outcome variables in the final model.

	Self-Infection Risk Perception	Threat Perception	Community	Ontario	Canada	World
Percieved household income	0.016	−0.087	−0.110 *	−0.061	−0.115 *	−0.024
Gender	−0.053	0.018	−0.115 *	−0.184 **	−0.196 **	−0.249 **
Education	−0.041	−0.152 *	−0.015	−0.049	−0.064	−0.033
Age	−0.083	0.029	−0.027	−0.058	−0.005	−0.010
Household size	−0.024	−0.133 **	0.115 *	−0.052	−0.063	0.000
Employmnet Status	−0.145 **	−0.004	−0.075	−0.022	−0.006	0.005
Life Satisfaction	−0.194 **	−0.148 **	−0.161 **	−0.126 *	−0.101	−0.082
Personal connection with COVID	0.229 **	0.111 *	0.199 **	0.198 **	0.233 **	0.206 **
Self-isolation expereince	−0.031	0.110 *	0.024	0.032	0.015	−0.013
Percieved anti-Chinese discrimination	0.235 **	0.239 **	0.182 **	0.193 **	0.179 **	0.162 **
Confusion over COVID-19 information	0.086 *	0.089 *	0.101 *	0.105 *	0.088	0.073

Note: * *p* < 0.05, ** *p* < 0.001.

## Data Availability

Data and the data analysis files can be retrieved from https://osf.io/df3kp/?view_only=aaa30b8b923f4a3bbcb964c602e0cc09 (uploaded on 28 October 2022).

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
