# Peer review of "Demographic and COVID Experience Predictors of COVID-19 Risk Perception among Chinese Residents in Canada"

_ijerph, 2022, doi:10.3390/ijerph192114448_

Round 1
Reviewer 1 Report
The article is clear and written appropriately. My great concern is that there is no novelty or innovation in this study, thus it does not have a substantial contribution to the readers.
The fact that women show higher infection rate predictions is expected, as throughout studies it (consistently) was shown that women have lower levels of reported resilience and higher levels of distress symptoms, Likewise, the association between exposure to people that contracted COVID and higher threat perceptions is expected and well-known.
Thus, I feel that at this advanced stage in which much is already known (and published) concerning the pandemic, this paper does not significantly add to the knowledge and insights of the readers. If the authors could study additional predictors, beyond demographics, this would be of contribution
Author Response
Dear Reviewer,
Thank you for the invitation to revise and resubmit this manuscript (ijerph-1967056). We substantially revised the manuscript to address your comments. All changes to the manuscript are in red text. We look forward to your further evaluation and potential publishing this paper in the International Journal of Environmental research and Public Health.
Concern:
The findings related to gender and exposure to COVID are expected and well known
RESPONSE: That is true! Many other researchers have found similar results. What was unique about our study in particular as the population measured, as well as how risk perception was measured. The novelty of this study has been more explicitly addressed in the text. “The novelty and unique value of this study stems from the focus on the currently under-researched Chinese Canadians, refined evaluation of different types of risk prediction (e.g., percentile infection rate prediction), and the consideration of comprehensive potential demographic and COVID experience predictors” (page 3).
Concern: The paper does not significantly add to the knowledge and insights of the readers. If the authors could study additional predictors, beyond demographics, this would be of contribution
RESPONSE: The results were re-analyzed after adding several new variables in order to address concerns over novelty. These variables included both demographic predictors (e.g., life satisfaction, employment), predictors unique to COVID-19 (self-isolation experience, confusion over COVID-19 information), and unique to our population of interest (perceived anti-Chinese discrimination). Additionally, dummy coding (see table 1 on page 4), was utilized to allow for meaningful interpretations of the categorical data (i.e., education, age, household size, employment). While the main results remained the same (gender as a predictor of future infection rate predictions, and personal connection to covid-19 as a predictor to all three measures of risk perception), there were several new significant predictors found (education, household size, employment, perceived anti-Chinese discrimination, and confusion over COVID-19 information; see page 6-7) The related tables, figures, and the discussion section were also updated to reflect this change in data analysis (see pages 4-9).
Reviewer 2 Report
This is an interesting topic on demographic predictors for COVID-19 risk perception 8 among Chinese residents in Canada. The authors found that gender and personal COVID exposure experience as the two strongest predictors. The study is well performed, including statistics and data interpretation. The discussion and limitations are also discussed sufficiently. I have nothing to add and suggest acceptation as is.
Author Response
Dear Reviewer,
Thank you thank you for your review this manuscript (ijerph-1967056), your comments were very much appreciated. Due to other feedback, we received, we substantially revised the manuscript. All changes to the manuscript are in red text. We look forward to your further evaluation and potential publishing this paper in the International Journal of Environmental research and Public Health.
Concern: No concerns raised
RESPONSE:
There were some concerns raised over the novelty of this manuscript, and in order to address this several new variables were included as predictors. For ease of reference, all changes to the manuscript are in red text. These new variables included both demographic predictors (e.g., life satisfaction, employment), predictors unique to COVID-19 (self-isolation experience, confusion over COVID-19 information), and unique to our population of interest (perceived anti-Chinese discrimination). Additionally, dummy coding (see table 1 on page 4), was utilized to allow for meaningful interpretations of the categorical data (i.e., education, age, household size, employment) While the main results remained the same (gender as a predictor of future infection rate predictions, and personal connection to covid-19 as a predictor to all three measures of risk perception), there were several new significant predictors found (education, household size, employment, perceived anti-Chinese discrimination, and confusion over COVID-19 information; see page 6-7) . The related tables and the discussion section were also updated to reflect this change in data analysis (see pages 4-9).
Additionally, the novelty of this study was more explicitly addressed in the text. “The novelty and unique value of this study stems from the focus on the currently under-researched Chinese Canadians, refined evaluation of different types of risk prediction (e.g., percentile infection rate prediction), and the consideration of comprehensive potential demographic and COVID experience predictors” (page 3).
Reviewer 3 Report
Dear authors,
I read your manuscript with interest as COVID-19 pandemic and its consequences are still a hot topic. However, i have some concerns about the article.
Important characteristics have not been included such as proffesion / work exposure charactristics. Word / profession / unemployment might be a quite important factor in the self-perceived risk whick cannot be depicted in the education and income characterisitcs (ex doctor, nurse, teacher--> great exposure? more informed?, working from home, not working--> less exposure?). Also, if a large snowball recruited percentage of the population were of similar profession bias could be even more and not easy to detect.
The importance of the article and its novelty are not well mentioned in the article. It is suggested that "promoting and messaging proper COVID-19 safety measures, specifically for men and those without a personal connection to COVID" might be the conclusion since these characteristics had lower self-perceived risk. I have concerns whether this extrapolation is safe. It might be safer to see if the level of self-perceived risk would reflect to different self-protecting attitudes, which would thus have a clinical significance and would prompt for recommendations.
It would be interesting to see if specific characteristics and self-perceived risk reflect to different levels of anxiety or quality of life. It would be also interesting to compare with other populations as the Chinese residents of Canada might have or not some differences from other populations. Finally, since the first wave a lot have changed. Could the vaccination have changed the perceived risk?
For these concerns i will have to reject your manuscript.
Best Regards.
Author Response
Dear Reviewer,
Thank you for the invitation to revise and resubmit this manuscript (ijerph-1967056). We substantially revised the manuscript to address your comments. All changes to the manuscript are in red text. Additionally, dummy coding (see table 1 on page 4), was utilized to allow for meaningful interpretations of the categorical data (i.e., education, age, household size, employment). After new variables were added, while the main results remained the same (gender as a predictor of future infection rate predictions, and personal connection to covid-19 as a predictor to all three measures of risk perception), there were several new significant predictors found (education, household size, employment, perceived anti-Chinese discrimination, and confusion over COVID-19 information). The related tables and the discussion section were also updated to reflect this change in data analysis. We look forward to your further evaluation and potential publishing this paper in the International Journal of Environmental research and Public Health.
Concern: Important characteristics have not been included; such as profession/work exposure characteristics (e.g., certain jobs may have higher exposure or be more informed).
RESPONSE:
This was an excellent suggestion for the data. We did originally collect data on profession, and have now included it as variable into our model. We dummy coded it (see table 1 on page 4), based on the most meaningful groups for our data, which resulted in us comparing those working in a medical field to those working in a non-medical field, and comparing those working in a medical field to those not currently employed. Unfortunately, the original question utilized a multiple-choice questionnaire, and in order to stream-line the question in the survey, there were limited options available for participants to select. Thus, we do not have data that distinguishes between the other groups you mention, those working as teachers, nor those working from home. However, we believe that the groups we were able to compare were the most meaningful groups for the data, as those working in the medical field would be expected to experience the most extreme example of exposure and being informed about the COVID-19 pandemic. (See results page 5-7).
Concern: Similar profession bias may exist in the sample due to the use of snowballing for data collection.
RESPONSE:
This is a very valid concern. We are happy to report that there was variability in our sample among employment. However, 43% did work in a non-medical field, and only 3% worked in a medical field (see table 1, page 4) We have addressed this as a limitation in the manuscript (page 9).
Concern: The importance of the article and its novelty are not well mentioned, and the extrapolation of the findings may not be safe.
RESPONSE:
The novelty of this study was more explicitly addressed in the text. The novelty and unique value of this study stems from the focus on the currently under-researched Chinese Canadians, refined evaluation of different types of risk prediction (e.g., percentile infection rate prediction), and the consideration of comprehensive potential demographic and COVID experience predictors” (page 3). We included additional predictor variables in order to accomplish this. Additionally, we re-addressed the importance of the results in the text and reclarified the conclusion of the findings.
Concern: It may be safer to see if the level of self-perceived risk would reflect in different self-protecting attitudes.
RESPONSE:
This was an excellent and exciting suggestion for the data. We had data from the original survey on self-isolation experience. Specifically, we asked participants if they had been self-isolating. What is fascinating here is that while almost exactly half (49%) of our sample reported self-isolating (thus showing variability in our sample, see table 1, page 4). (See results page 5-7).
Additionally, we also included the predictor Confusion over COVID-19 information, in order to see if this may be an element related to risk perception. (See results page 5-7).
Concern: It would be interesting to see how anxiety and quality of life relate to self-risk perception and different characteristics.
We agree! In fact, we have a published paper that directly addressed how anxiety (in addition to depression, and stress) are related to different characteristics and risk perception measures. However, since we have already reported this data from this specific data set, we could not include this data in this manuscript. (https://www.ncbi.nlm.nih.gov/pmc/articles/PMC8750305/)
However, quality of life was not addressed in our previous manuscript. We did collect data on life satisfaction, and included that in the model in this current manuscript to see how this would relate to our different measures of risk perception, when controlling for other predictors (see figure 1, page 7). We however believed that an analysis into the characteristics that may be causing variances in life satisfaction was beyond the scope of this manuscript, as this manuscript is exclusively focused on predictors of risk perception.
Concern: It would be interesting to see other populations, and changes since the first wave including the introduction of vaccines.
We agree, both of these items would be very interesting to research! However, both of these items are unfortunately beyond the scope of this current manuscript. That being said, we agree that this would be an excellent next step, and have included them in our suggestions for future research (page 9).
However, in order to encapsulate why Chinese-Canadians were a unique group to study, particularly during the first wave, we included another predictor into the model that was unique to the stressors that our population was experiencing. Specifically, perceived anti-Chinese discrimination was included as a predictor (see results page 5-7).
Round 2
Reviewer 1 Report
Thank you for revising the article. It is publishable in its present form
Reviewer 3 Report
Dear authors,
I read your revised manuscript with great interest. I find the revisions satisfactory and i change the decision to "Accept".
Best regards.